# A Fair Classifier Using Kernel Density Estimation

**Jaewoong Cho**
EE, KAIST
cjw2525@kaist.ac.kr

**Gyeongjo Hwang**
EE, KAIST
hkj4276@kaist.ac.kr

**Changho Suh**
EE, KAIST
chsuh@kaist.ac.kr

## Abstract

As machine learning becomes prevalent in a widening array of sensitive applications such as job hiring and criminal justice, one critical aspect in the design of machine learning classifiers is to ensure fairness: Guaranteeing the irrelevancy of a prediction to sensitive attributes such as gender and race. This work develops a kernel density estimation (KDE) methodology to faithfully respect the fairness constraint while yielding a tractable optimization problem that comes with high accuracy-fairness tradeoff. One key feature of this approach is that the fairness measure quantified based on KDE can be expressed as a differentiable function w.r.t. model parameters, thereby enabling the use of prominent gradient descent to readily solve an interested optimization problem. This work focuses on classification tasks and two well-known measures of group fairness: demographic parity and equalized odds. We empirically show that our algorithm achieves greater or comparable performances against prior fair classifers in accuracy-fairness tradeoff as well as in training stability on both synthetic and benchmark real datasets.

## 1 Introduction

During the last decade, we have witnessed an unprecedented explosion of academic and popular interests in machine learning. Machine learning is no longer just an engine behind image classifiers and spam filters. It is now employed to make critical decisions that affect our lives, cultures, and rights, e.g., screening job applicants, and informing bail & parole decisions. With a surge of such applications, one major criterion in the design of machine learning algorithms is to ensure *fairness*.

A fair classifier aims at achieving the irrelevancy of a prediction to sensitive attributes such as race, sex, age, and religion. Prior works in the fairness literature have developed several metrics that capture various notions of discrimination. Three major fairness measures have been taken into consideration: (i) *group* fairness [8, 3, 12, 42, 41] that intends to ensure similar statistics across distinct demographics; (ii) *individual* fairness [7, 9, 33, 40] that targets nondiscriminatory predictions across nearby examples; (iii) *causality-based* fairness counterparts [20, 24, 30, 37, 45, 46, 18]. This work focuses on group fairness that has been widely explored in a variety of applications. Prominent group fairness measures include demographic parity [8, 42], equal opportunity [12], and equalized odds [12]. All of these intend to quantify how prediction outputs vary depending on sensitive attributes.

There has been a proliferation of fair classifiers [12, 7, 5, 1, 28, 43, 17, 23]. One challenge that arises in the prior algorithms is that they suffer from obtaining an explicit and possibly differentiable fairness measure w.r.t. model parameters and the non-differentiability often prevents the use of popular algorithms such as gradient descent. This naturally leads to a common approach: Incorporating an *expressible fairness proxy* as a regularization term in an interested optimization. One pioneering work along this direction [42] employs as a fairness proxy a covariance function between a sensitive attribute and a prediction. However, such a proxy-based approach may not well respect fairness constraints when it serves as a *weak* constraint as in [42]. A small covariance does not ensure *statistical independence* although the reverse is always the case. Hence, any theoretical performance

is not guaranteed for a wide range of real datasets in which the low correlation may not necessarily ensure independence.

**Contribution:** To address the issue, we take a distinct approach that allows us to directly quantify fairness measures without relying on such proxy. Our methodology is based on kernel density estimation (KDE) [4] that serves to estimate a probability distribution. We emphasize three notable aspects of our KDE-based framework. The first is that it enables a *direct* computation of an interested fairness measure without introducing any proxy. Second, it yields high accuracy on the distribution estimate. In the binary classifier of our consideration, a moderate sample size ensures a reasonably precise estimate, in stark contrast to high-dimensional settings [34, 32, 14]; see Remark 1 for details. Lastly, the fairness measure computed based on KDE can be expressed as a *differentiable* function w.r.t. model parameters, thereby enabling the use of standard gradient descent to easily solve a constrained optimization problem taking the fairness measure as a regularization term. Our extensive experiments conducted both on synthetic and benchmark real datasets (Law School Admissions [36], Adult Census [6], Credit Card Default [6, 39], and COMPAS [2]) demonstrate that our algorithm achieves higher accuracy-fairness tradeoff relative to the states of the arts [42, 41, 44, 1, 25, 12], both w.r.t. demographic parity and equalized odds. It also exhibits an enhanced performance in training stability, compared to adversarial learning based frameworks [44, 11].

**Related works:** Fair classifiers are categorized broadly into the following three types: (1) *pre*-processing; (2) *post*-processing; (3) *in*-processing. Pre-processing intends to correct biased and/or possibly poisoned data (if any) for mitigating discrimination [16, 43, 8, 38] while post-processing perturbs classifier's output at test time while freezing the model [12, 28]. In-processing handles a fairness constraint in the process of model training. Below we provide a list of in-processing techniques most relevant to ours.

One common in-processing approach is to address a constrained optimization that incorporates a fairness measure as a regularization term. Zafar et al. [42] takes this approach, yet utilizing a *covariance fairness proxy* w.r.t. a prediction and a sensitive attribute. While such covariance proxy yields convex optimization under the logistic regression and SVM frameworks to achieve the optimality via gradient descent, it comes at a cost in enforcing a fairness constraint, as it serves only as a weak constraint. Other approaches based on linear regression and SVM include [17, 5].

Another line of in-processing algorithms which yet take different approaches are [44, 1, 25, 29]. Zhang et al. [44] build upon an adversarial learning framework [11] to design a classifier and a discriminator so that the discriminator cannot identify a sensitive attribute from a prediction. While it may enjoy promising accuracy-fairness tradeoff with careful design, it suffers from a stability issue in training as it is based on min-max optimization [11, 31]. See Section 4 for a relevant in-depth discussion.

## 2 Problem Formulation

A fair classifier setting includes two types of data: (i) normal (and possibly objective) data; (ii) *sensitive* data (or called *sensitive attributes*). We denote the normal data by $x \in \mathbb{R}^d$. In the case of recidivism score prediction [2], such $x$ may refer to a collection of the number of prior criminal records and a criminal type, e.g., misdemeanor or felony. For *sensitive* data, we employ a different notation, say $z$. In the above example, $z$ may indicate a race type among white ($z = 1$) and black ($z = 0$). In general, the alphabet size of $z$ is arbitrary. There are many race types such as Black, White, Asian, Hispanic, to name a few. Also, there could be multiple sensitive attributes like gender and race (e.g., White male, White female, Black male, Black female). In order to reflect such scenarios, we consider discrete-valued $z \in \mathcal{Z}$ with an arbitrary alphabet size $|\mathcal{Z}|$. Let $\hat{y}$ be the classifier output which aims to represent the ground-truth conditional distribution $p(y|x,z)$. Here $y \in \mathcal{Y}$ denotes the ground-truth label. In the recidivism score prediction, $y = 1$ means reoffending in the near future, say within two years ($y = 0$ otherwise), while $\hat{y}$ indicates the probability of such event occurring. We are given $m$ example triplets: $\{(x^{(i)}, z^{(i)}, y^{(i)})\}_{i=1}^{m}$. We assume that both $x$ and $z$ are fed as the input, although $z$ may not be part of the input in an effort to automatically respect disparate treatment [42], another fairness notion that captures an unequal treatment. For a clearer explanation, we first focus a binary classification setting where $\mathcal{Y} = \{0, 1\}$ and then discuss on a multiclass setting as presented in Section 5.

This work focuses on two group fairness notions: demographic parity and equalized odds [8, 42, 12]. Their formal definitions rely on a few notations. Let $Z \in \mathcal{Z}$ be a random variable that indicates a sensitive attribute. Let $\widetilde{Y} \in \mathcal{Y}$ be a hard-decision value of the prediction $\widehat{Y}$ at a certain threshold: $\widetilde{Y} := \mathbf{1}\{\widehat{Y} \geq \tau\}$ where $\tau \in [0, 1]$.

**Definition 1 (Demographic Parity)** *A classifier is said to satisfy demographic parity if its prediction* $\widetilde{Y}$ *is independent of the sensitive attribute* $Z$: $\Pr(\widetilde{Y} = 1|Z = z) = \Pr(\widetilde{Y} = 1)$, $\forall z \in \mathcal{Z}$.

One popular measure that captures the degree of violating demographic parity is the difference between the conditional probability and its marginal. Similar to the namings in [5, 15], we call it the difference w.r.t. demographic parity (DDP):

$$\mathsf{DDP} := \sum_{z \in \mathcal{Z}} |\Pr(\widetilde{Y} = 1|Z = z) - \Pr(\widetilde{Y} = 1)| \tag{1}$$

where $\mathsf{DDP} \geq 0$ and the equality holds when a classifier fully respects demographic parity. One may consider another measure which takes "$\max$" operation instead of "$\sum$" in (1), or a different measure, called *disparate impact*, which captures the degree of the statistical independence via the ratio of probabilities of positive events among distinct sensitive attributes [1, 8, 42]. We focus on $\mathsf{DDP}$ in (1) for tractability of an associated optimization problem that we will detail later.

**Definition 2 (Equalized Odds)** *A classifier is said to satisfy equalized odds if its prediction is conditionally independent of the sensitive attribute* $Z$ *given the label* $Y$: $\Pr(\widetilde{Y} = 1|Z = z, Y = y) = \Pr(\widetilde{Y} = 1|Y = y)$, $\forall z \in \mathcal{Z}$ *and* $y \in \{0, 1\}$.

Similarly we define the difference of the two probability quantities w.r.t. equalized odds (DEO) as:

$$\mathsf{DEO} := \sum_{y \in \{0,1\}} \sum_{z \in \mathcal{Z}} |\Pr(\widetilde{Y} = 1|Z = z, Y = y) - \Pr(\widetilde{Y} = 1|Y = y)| \tag{2}$$

where $\mathsf{DEO} \geq 0$ and the equality holds when a classifier respects equalized odds.

One natural approach to decrease $\mathsf{DDP}$ or $\mathsf{DEO}$ is to incorporate the fairness-related constraint as a *regularization* term into the conventional optimization which is often of the following form:

$$\min_{w} \frac{1}{m} \sum_{i=1}^{m} \ell_{\mathsf{CE}}(y^{(i)}, \hat{y}^{(i)}) \tag{3}$$

where $\ell_{\mathsf{CE}}(y, \hat{y}) := -\sum_{j} y_j \log \hat{y}_j$ indicates cross entropy loss [10], and $w$ denotes weights of a classifier. Taking into account $\mathsf{DDP}$ or $\mathsf{DEO}$, we then obtain:

$$\min_{w} \frac{1 - \lambda}{m} \sum_{i=1}^{m} \ell_{\mathsf{CE}}(y^{(i)}, \hat{y}^{(i)}) + \lambda \mathcal{L}_{\mathsf{fair}} \tag{4}$$

where the fairness-associated regularization term $\mathcal{L}_{\mathsf{fair}}$ takes $\mathsf{DDP}$ or $\mathsf{DEO}$, and $\lambda \in [0, 1]$ denotes a normalized regularization factor that balances predication accuracy against the fairness constraint. Here one challenge that arises is that expressing $\mathsf{DDP}$ and $\mathsf{DEO}$ in terms of $w$ is not that straightforward. One effort was made by [42] which introduced a *surrogate yet expressible* fairness measure. Specifically they employ a covariance function between $\widehat{Y}$ and $Z$ as a fairness proxy. However, this covariance proxy serves only as a weak constraint and therefore it may not fully respect the fairness constraint.

To overcome the challenge, we take a kernel density estimation (KDE) [4] trick which allows us to *faithfully* quantify fairness measures. Our approach also enables the computed measures to be differentiable w.r.t. $w$, thus enjoying a variety of gradient-based optimizers [10, 19].

## 3  Proposed Approach

The computations of $\mathsf{DDP}$ and $\mathsf{DEO}$ require the knowledge of $P_{\widetilde{Y}|Z}$ and $P_{\widetilde{Y}|Z,Y}$, respectively. So the question of interest boils down to: How to express $P_{\widetilde{Y}|Z}$ and $P_{\widetilde{Y}|Z,Y}$ in terms of $w$? To this end, we

employ the KDE methodology that serves to estimate the pdf of $\widehat{Y}$ via samples. Since $\widetilde{Y}$ is a function of $\widehat{Y}$, the pmf of $\widetilde{Y}$ can be represented also via the samples. A notable aspect of the KDE approach (to be detailed in the sequel) is that it enables $P_{\widetilde{Y}|Z}$ and $P_{\widetilde{Y}|Z,Y}$ to be expressed as *differentiable* functions w.r.t. $w$. Let us start by reviewing the KDE.

**Definition 3 (Kernel Density Estimator (KDE) [4])** *Let $(\hat{y}^{(1)}, \ldots, \hat{y}^{(m)})$ be i.i.d. examples drawn from a distribution with an unknown density $f$. Its KDE is defined as:*

$$\hat{f}(\hat{y}) := \frac{1}{mh} \sum_{i=1}^{m} f_k \left( \frac{\hat{y} - \hat{y}^{(i)}}{h} \right) \tag{5}$$

*where $f_k$ is a kernel function (see Definition 4) and $h > 0$ is a smoothing parameter called bandwidth.*

**Definition 4 (A kernel function)** *A kernel function is a non-negative real-valued integrable function $f_k(\cdot)$ that satisfies two requirements: normalization and symmetry.*

Here we employ a prominent Gaussian kernel function:

$$f_k(\hat{y}) := \frac{1}{\sqrt{2\pi}} \exp \left( -\frac{\hat{y}^2}{2} \right). \tag{6}$$

To ease the computation of the cdf of $f_k(\hat{y})$, we approximate the $Q$-function as per [21]:

$$F_k(\hat{y}) := \int_{\hat{y}}^{\infty} f_k(y) dy = Q(\hat{y}) \approx e^{-a\hat{y}^2 - b\hat{y} - c} \tag{7}$$

where $(a, b, c) = (0.4920, 0.2887, 1.1893)$.

## 3.1 Demographic Parity

We first estimate $f_{\widehat{Y}|Z}(\hat{y}|z)$ using the KDE:

$$\hat{f}_{\widehat{Y}|Z}(\hat{y}|z) = \frac{1}{m_z h} \sum_{i \in I_z} f_k \left( \frac{\hat{y} - \hat{y}^{(i)}}{h} \right) \tag{8}$$

where $I_z := \{i : z^{(i)} = z\}$ and $m_z := |I_z|$. This together with $\widetilde{Y} := \mathbf{1}\{\widehat{Y} \geq \tau\}$ gives:

$$\hat{P}_{\widetilde{Y}|Z}(1|z) = \int_{\tau}^{\infty} \hat{f}_{\widehat{Y}|Z}(\hat{y}|z) d\hat{y} = \frac{1}{m_z} \sum_{i \in I_z} F_k \left( \frac{\tau - \hat{y}^{(i)}}{h} \right)$$

where $F_k(\hat{y}) := \int_{\hat{y}}^{\infty} f_k(y) dy$.

**Proposition 1** *Since $f_k(\hat{y})$ is continuous and each $\hat{y}^{(i)}$ is a differentiable function w.r.t. $w$, $\hat{P}_{\widetilde{Y}|Z}$ is also differentiable. Using the chain rule, one can then compute its gradient as:*

$$\nabla_w \hat{P}_{\widetilde{Y}|Z}(1|z) = \frac{1}{m_z h} \sum_{i \in I_z} f_k \left( \frac{\tau - \hat{y}^{(i)}}{h} \right) \cdot \nabla_w \hat{y}^{(i)}. \tag{9}$$

Now let us consider the DDP constrained optimization:

$$\min_{w} \frac{1 - \lambda}{m} \sum_{i=1}^{m} \ell_{\mathsf{CE}}(y^{(i)}, \hat{y}^{(i)}) + \lambda \cdot \mathsf{DDP} \tag{10}$$

where

$$\mathsf{DDP} \approx \sum_{z \in \mathcal{Z}} \left| \hat{P}_{\widetilde{Y}|Z}(1|z) - \hat{P}_{\widetilde{Y}}(1) \right| \text{ and } \hat{P}_{\widetilde{Y}}(1) = \sum_{z \in \mathcal{Z}} \frac{m_z}{m} \hat{P}_{\widetilde{Y}|Z}(1|z). \tag{11}$$

For tractability of the non-differentiable absolute function $|\cdot|$, we employ the Huber loss [13]:

$$\mathsf{DDP} \approx \sum_{z \in \mathcal{Z}} H_\delta \left( \hat{P}_{\widetilde{Y}|Z}(1|z) - \hat{P}_{\widetilde{Y}}(1) \right) \text{ where } H_\delta(x) := \left\{ \begin{array}{ll} \frac{1}{2}x^2 & \text{for } |x| \le \delta; \\ \delta(|x| - \frac{1}{2}\delta) & \text{otherwise.} \end{array} \right.$$

This together with (9) and (11) yields an approximation of the gradient of $\mathsf{DDP}$:

$$\nabla_w \mathsf{DDP} \approx \sum_{z \in \mathcal{Z}} H'_\delta \left( \hat{P}_{\widetilde{Y}|Z}(1|z) - \hat{P}_{\widetilde{Y}}(1) \right) \cdot \nabla_w \left( \hat{P}_{\widetilde{Y}|Z}(1|z) - \hat{P}_{\widetilde{Y}}(1) \right). \tag{12}$$

We employ a neural network (NN) for $w$ and gradient descent for training the fair classifier (10). For a linear classifier, we can indeed compute the gradient explicitly, so we provide the closed-form of the gradient in the supplementary. For general multi-layer NNs, on the other hand, an explicit formula is rather messy to express, while it can readily be implemented with autograd under machine learning frameworks. Hence, we do not leave the detailed formula for general NNs.

### 3.2 Equalized Odds

Taking the KDE approach, similarly we obtain:

$$\hat{P}_{\widetilde{Y}|Z,Y}(1|z,y) = \frac{1}{m_{zy}} \sum_{i \in I_{zy}} F_k \left( \frac{\tau - \hat{y}^{(i)}}{h} \right) \tag{13}$$

where $I_{zy} := \{i : z^{(i)} = z, y^{(i)} = y\}$ and $m_{zy} := |I_{zy}|$. We can then compute the gradient w.r.t. $w$:

$$\nabla_w \hat{P}_{\widetilde{Y}|Z,Y}(1|z,y) = \frac{1}{m_{zy}h} \sum_{i \in I_{zy}} f_k \left( \frac{\tau - \hat{y}^{(i)}}{h} \right) \cdot \nabla_w \hat{y}^{(i)}. \tag{14}$$

Now consider the $\mathsf{DEO}$ constrained optimization problem:

$$\min_w \frac{1-\lambda}{m} \sum_{i=1}^{m} \ell_{\mathsf{CE}}(y^{(i)}, \hat{y}^{(i)}) + \lambda \cdot \mathsf{DEO}. \tag{15}$$

Again using the KDE together with the Huber loss, we approximate:

$$\mathsf{DEO} \approx \sum_{y \in \{0,1\}} \sum_{z \in \mathcal{Z}} H_\delta \left( \hat{P}_{\widetilde{Y}|Z,Y}(1|z,y) - \hat{P}_{\widetilde{Y}|Y}(1|y) \right) \tag{16}$$

where $\hat{P}_{\widetilde{Y}|Y}(1|y) = \sum_{z \in \mathcal{Z}} \frac{m_{zy}}{m_y} \hat{P}_{\widetilde{Y}|Z,Y}(1|z,y)$ and $m_y := |\{i : y^{(i)} = y\}|$. This then yields:

$$\nabla_w \mathsf{DEO} \approx \sum_{y \in \{0,1\}} \sum_{z \in \mathcal{Z}} H'_\delta \left( \hat{P}_{\widetilde{Y}|Z,Y}(1|z,y) - \hat{P}_{\widetilde{Y}|Y}(1|y) \right) \cdot \nabla_w \left( \hat{P}_{\widetilde{Y}|Z,Y}(1|z,y) - \hat{P}_{\widetilde{Y}|Y}(1|y) \right).$$
$$\tag{17}$$

Again we employ an NN for $w$ and provide the explicit gradient formula for a linear classifier in the supplementary, while omitting complicated gradient expressions for general NNs. We use gradient descent for training the fair classifier (15).

**Remark 1 (Estimate accuracy of the KDE approach)** *In general, the KDE approach yields an inaccurate distribution estimate under* high-dimensional *settings with a moderate amount of samples, as the sample size required for a reasonably good estimate should scale exponentially with the dimension [34, 32, 14]. However, this is not the case in our setting that emphasizes the binary classifier. In our binary classifier setting $\hat{y} \in \mathbb{R}$, the required sample size is not prohibitively large even for a highly accurate estimate.* ∎

**Remark 2 (On the choice of the bandwidth $h$)** *While it is crucial to make a careful choice on $h$ for an accurate pdf estimate of $\widehat{Y}$, it is not the case in our setting which targets only a "pmf" estimate of a hard-decision value $\widetilde{Y}$. In fact, we find via experiments that a rough choice on $h$ suffices to yield*

*a good enough estimate of the pmf. The left plot in Fig. 1 shows a high sensitivity of the estimated pdf of $\widehat{Y}$ w.r.t. different $h$'s, while the right table demonstrates the robustness of the estimated pmf of $\widetilde{Y}(:= \mathbf{1}\{\widehat{Y} > 0.5\})$ against various $h$'s. Nevertheless, a more accurate "pdf" estimate based on an in-depth analysis revealing the bias-variance tradeoff [34, 35] would definitely yield a better pmf estimate, thereby leading to enhanced performance. Hence, we conduct such theoretical analysis and provide an in-depth discussion on the choice of $h$ in the supplementary.* ∎

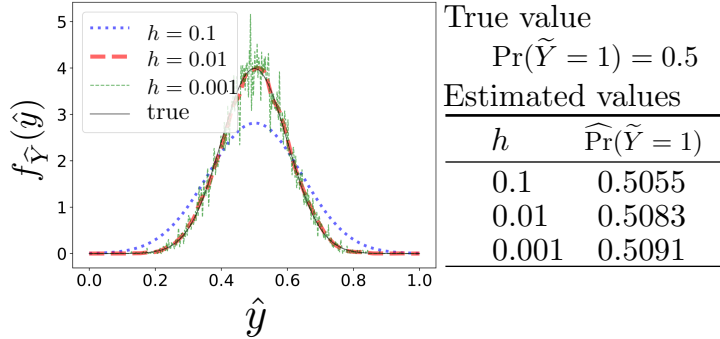

Figure 1: (Left) The pdf estimates of $\widehat{Y}$ via KDE using 10,000 examples from the normal distribution $\mathcal{N}(0.5, 1)$ for different bandwidths $h \in \{0.1, 0.01, 0.001\}$; (Right) The pmf estimates of $\widetilde{Y}(:= \mathbf{1}\{\widehat{Y} > 0.5\})$ via KDE for different $h$'s; (Implication) Relative to the pdf estimate of $\widehat{Y}$, the pmf estimate of $\widetilde{Y}$ is much more robust to a choice of $h$.

**Remark 3 (Faithful implementation of fairness measures & beneficial consequences)** *One key benefit of our approach is that it enables direct computation of interested fairness measures without relying on any fairness proxies such as covariance function [42]. Hence, we can ease training with standard gradient-based optimizers such as stochastic gradient descent (SGD) and Adam optimizer. We conduct extensive experiments on synthetic and benchmark real datasets to demonstrate that: (1) our algorithm outperforms prior fair classifiers in tradeoff performance both w.r.t. DDP and DEO; (2) it ensures stability in training unlike adversarial learning approaches; (3) the performance of our algorithm is robust to a choice of hyperparameters employed in the approach.* ∎

## 4 Experiments

We provide experimental results conducted on synthetic and four benchmark real datasets (COMPAS [2], Adult Census [6], Law School Admissions [36], and Credit Card Default [6, 39]). We implement our algorithm in PyTorch [26], and all experiments are performed on a server with GeForce GTX 1080 Ti GPUs. All of our results are on a separate test set.

### 4.1 Synthetic Dataset

We employ a simple yet non-trivial dataset (called the Moon dataset [27]) in which data are not linearly separable. See the left figure in Fig. 2. We consider a setting in which $x$ has two non-sensitive attributes (say $x_1$ and $x_2$), $z$ is ternary, and $y$ is binary (say $y = 1$ for a positive outcome; $y = 0$ otherwise). We leave a more detailed explanation of synthetic data generation in the supplementary. The dataset includes 15,000 examples, which are then split into two subsets: 80% train set ($m_{\text{train}} = 12,000$) and 20% test set ($m_{\text{test}} = 3,000$). We train fair classifiers with a 2-layer NN with 16 hidden nodes. For our approach, we set hyperparameters $\delta$ (of the Huber function) and $h$ to be 1 and 0.1, respectively. A theoretical insight on the choice of $h$ is provided in the supplementary. We use the batch size of 512. We use Adam optimizer and its default parameters $(\beta_1, \beta_2) = (0.9, 0.999)$ with the learning rate of $10^{-2}$.

Fig. 2 demonstrates accuracy-fairness tradeoff w.r.t. DDP, evaluated on the synthetic test set. In our approach, we sweep the tuning knob $\lambda$ from 0 and 1. Here each point corresponds to a particular $\lambda$ and it represents an average value over 5 trials with different seeds in training. We observe that our

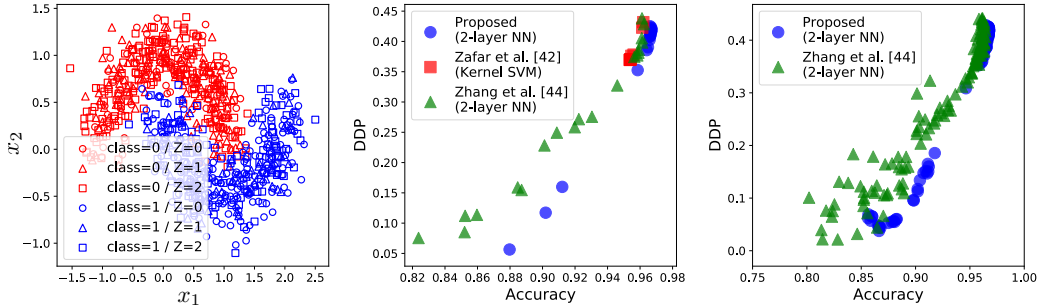

Figure 2: (Left) Visualization of our synthetic dataset; (Middle) Accuracy-fairness tradeoff w.r.t. demographic parity on the synthetic dataset; (Right) Accuracy-fairness tradeoff w.r.t. equalized odds on the synthetic dataset

algorithm outperforms by respectful margins the other baselines: (i) Zafar et al. [42] (a covariance-fairness-proxy-based classifier); (ii) Zhang et al. [44] (a GAN-based fair classifier). We also see that a covariance-based algorithm cannot achieve DDP performance below a certain level. We expect this may be because a small covariance does not necessarily guarantee statistical independence.

The right figure of Fig. 2 exhibits training stability. We compare ours to a state-of-the-art adversarial learning approach [44]. Here each point represents a performance evaluated only on a particular seed in training. So the variability of different points captures the degree of stability in training; the more dispersed, the more unstable. We see that [44] yields more spread points, relative to ours, meaning that our algorithm is more stable.

## 4.2 Real Datasets

We employ four benchmark real datasets: COMPAS [2], Adult Census [6], Law School Admissions [36], and Credit Card Default [6, 39]:

- *COMPAS*: The associated task is to predict the recidivism of criminals. We use 3,536 train examples and 1,742 test examples[1]. We construct $x$ with 8 normal features (age, criminal history, and more) and choose "race" (white vs. non-white) and "sex" for $z$.

- *Adult Census*: The associated task is to predict whether the annual income of an individual is above \$50,000. This dataset provides $m_{\text{train}} = 32,561$ and $m_{\text{test}} = 12,661$ examples [6]. We pre-process the data in the same way as done in [22][2] with "white-vs-non-white race" and "gender" as sensitive attributes.

- *Law School Admissions*: The task of interest is to predict whether an applicant gets an admission from a law school. Normal features include LSAT score, undergraduate GPA, gender and more, while the sensitive attribute is "white-vs-non-white race" [36]. We split the data into two subsets with $m_{\text{train}} = 77,267$ and $m_{\text{test}} = 19,317$.

- *Credit Card Default*: The target task is to predict whether a credit card user declares a default in the coming month. Age, education, marriage, and prior payment records are included in $x$, while $z$ refers to gender. We use 20% ($m_{\text{test}} = 6,000$) as a test set out of total $m = 30,000$ examples.

Note that we consider multiple sensitive attributes ($|\mathcal{Z}| = 4$) on COMPAS and Adult Census, while we use only one sensitive attribute for the others. For comparison, we consider five baselines: (i) Zafar et al. [42]; (ii) Zhang et al. [44]; (iii) Agarwal et al. [1]; (iv) Hardt et al. [12]; (v) Narasimhan [25]. We employ a 2-layer NN with 16 hidden nodes for all the baselines except for Zafar et al. [42] and Narasimhan [25], as the algorithms in [42] and [25] rely on convex optimization. More specifically, we use a kernel SVM for [42] and logistic regression for [25]. For a discriminator in Zhang et al. [44], we use a 2-layer NN (with 16 hidden nodes) on datasets with multiple sensitive attributes (COMPAS

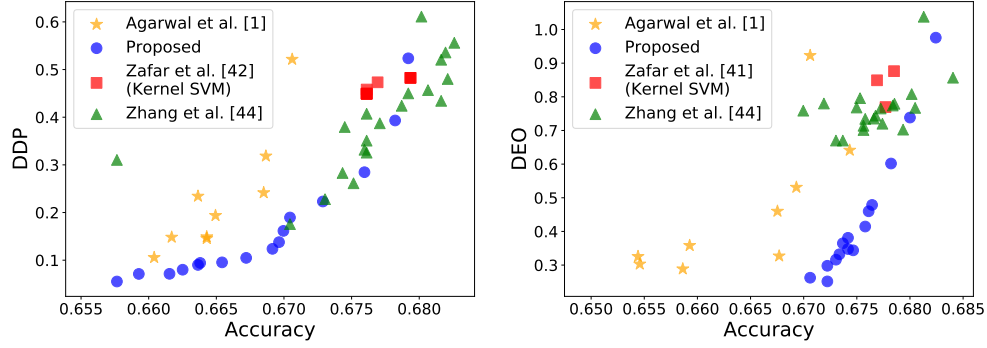

Figure 3: (Left) Accuracy-fairness tradeoff w.r.t. Demographic Parity evaluated on the COMPAS dataset; (Right) Equalized Odds counterpart.

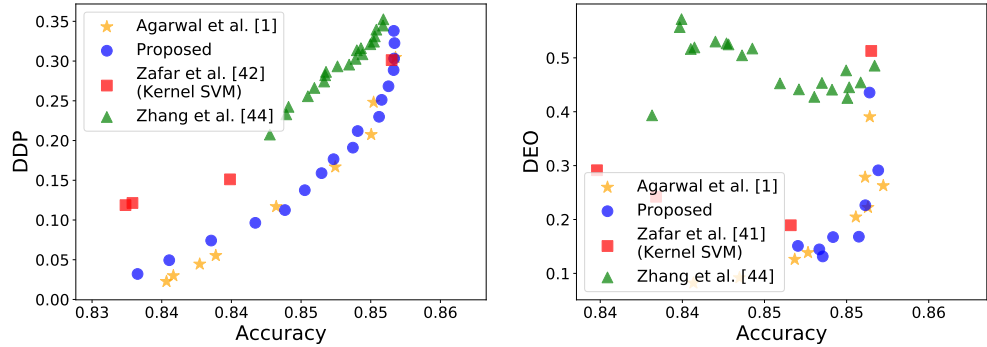

Figure 4: (Left) Accuracy-fairness tradeoff w.r.t. Demographic Parity evaluated on the Adult Census Income dataset; (Right) Equalized Odds counterpart.

and Adult Census) and linear on the other datasets. The left plot in Fig. 3 shows accuracy-fairness tradeoff w.r.t. DDP evaluated on COMPAS. Each point corresponds to a particular tuning knob and it represents an average value over 5 trials with different random seeds. Off-the-scale curves for low-performance baselines are not shown. We observe in the plot that the proposed algorithm achieves the best tradeoff. We also obtain similar results w.r.t. equalized odds as in the right plot of Fig. 3, except that the baseline [42] is replaced by [41]. We obtain the result that the proposed algorithm achieves near best tradeoff while being comparable to Agarwal et al. [1] on Adult Census. Another aspect, not reflected here in the plot, yet which we wish to emphasize, is computational complexity. Since Agarwal et al. [1] require *multiple rounds* of training, their algorithm exhibits significant computational time, which is much higher relative to ours. Explicit running times are reported in the supplementary. We also obtain similar results on the other two datasets: Law Shcool Admissions and Credit Card Default which leave in the supplementary.

## 5 Extension to multiclass classification

We now present a simple extension to multiclass setting where each element $\widehat{Y}_i$ of the softmax output $\widehat{Y} \in \mathbb{R}^{|\mathcal{Y}|}$ can be interpreted as $\Pr(Y = i | X = x, Z = z)$. A hard decision value of the prediction is determined by: $\widetilde{Y} = \arg\max_i \widehat{Y}_i$. A multiclass classifier satisfies demographic parity if $P_{\widetilde{Y}}(y) = P_{\widetilde{Y}|Z}(y|z), \forall y \in \mathcal{Y}$ and $\forall z \in \mathcal{Z}$. This naturally leads to the following form of DDP:

$$\text{DDP} := \sum_y \sum_z |\Pr(\widetilde{Y} = y | Z = z) - \Pr(\widetilde{Y} = y)|. \tag{18}$$

By using the fact that

$$\widehat{Y}_j > 0.5 \Rightarrow \arg\max_i \widehat{Y}_i = j, \qquad (19)$$

we estimate $\Pr(\widetilde{Y} = y | Z = z)$ and $\Pr(\widetilde{Y} = y)$ as a differentiable function of $w$:

$$\widehat{P}_{\widetilde{Y}|Z}(y|z) \approx \Pr(\hat{Y}_j > 0.5 | Z = z) = \int_{0.5}^{\infty} \hat{f}_{\widehat{Y}_j|Z}(\hat{y}|z) d\hat{y} = \frac{1}{m_z} \sum_{i \in I_z} F_k \left( \frac{\tau - \hat{y}_j^{(i)}}{h} \right). \qquad (20)$$

Then similarly with the binary case in (12), we can obtain the gradient $\nabla_w \mathsf{DDP}$. We provide experimental results of such extension on a synthetic dataset with three classes. We leave a more detailed explanation of this dataset in the supplementary. We train fair classifiers with a 2-layer NN with 16 hidden nodes. We observe that our simple extension offers respectful margins especially in the perfect fairness regime. For comparison, we consider the One-vs-all extension of (i) Zafar et al. [42] w/ Kernel SVMs and (ii) Zhang et al. [44] that naturally extends to the multiclass setting.

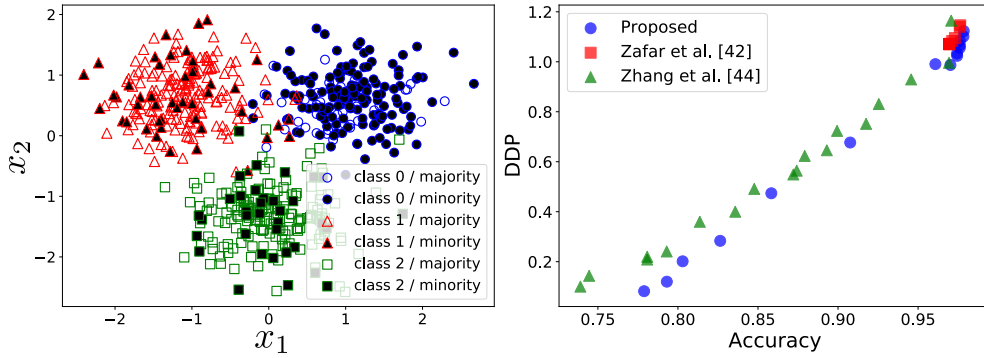

Figure 5: (Left) Visualization of 3-way Gaussian synthetic dataset; (Right) Accuracy-fairness trade-off w.r.t demographic parity with a 2-layer NN.

# 6 Conclusion

We proposed a computationally efficient KDE-based fair classifier under a binary classification setting that achieves the best accuracy-fairness tradeoff while enjoying the standard gradient-based optimizers via obtaining a differentiable cost function. The proposed algorithm also yields an improved performance in training stability that GAN-based fair classifiers suffer from. The beneficial aspects of our algorithm are well presented via a variety of synthetic and real-data experiments w.r.t. two major group fairness measures: demographic parity and equalized odds. One future work of interest is to extend to other fairness notions such as individual fairness and causality-based fairness.

## Broader Impact

The optimality-efficiency-stability aspects of our algorithm will offer an opportunity to replace the current fair classifiers which either are far from optimality, entail high complexity, and/or suffer from the stability issue. In particular, our optimization framework will play a role in stabilizing the training, which many of the GAN-based fair classifiers suffer from. Hence, it can give significant impacts upon a widening array of machine learning systems that have relied upon GAN-based architectures, being powerful in a wide variety of applications.

The current KDE approach is tailored for the binary classifier setting. Hence, a naive extension to general multiclass classifiers (high-dimensional settings) might incur an inaccurate estimate of an interested distribution, thus potentially exhibiting a poor accuracy-fairness tradeoff. Another flip side lies in its robustness against adversarial attacks. It was recently reported in [29] that existing fair classifiers are vulnerable to biased and/or poisoned data. An initial effort has also been made in the same paper to address both fairness and robustness issues. Hence, one future work of great potential might be to gracefully merge the idea in [29] with ours, potentially together with a non-straightforward extension to multiclass classifier settings.

## Acknowledgement

This work was supported by Institute for Information & communications Technology Planning & Evaluation(IITP) grant funded by the Korea government(MSIT) (No. 2019-0-01396, Development of framework for analyzing, detecting, mitigating of bias in AI model and training data)

## Footnotes

[1]https://github.com/microsoft/tempeh

[2]https://github.com/slundberg/shap

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
