[Supplementary Material · NeurIPS2020_Fairness__Camera_ready_supplementary.pdf]

# Supplementary Material for:
# A Fair Classifier Using Kernel Density Estimation

## 1  Outline

We first present a theorem that forms the basis of our choice made on the bandwidth $h$, as mentioned in Remark 2. Next we provide the gradient formulas that we explicitly derived for linear and 2-layer NN classifiers both w.r.t. DDP and DEO. We provide details on synthetic datasets and we provide experimental results conducted on two additional real benchmark datasets: Law School Admissions [36] and Credit Card Default [6, 39].

## 2  Theoretical Insights on the Choice of the Bandwidth $h$

We provide theoretical insights on the choice on $h$ such that it minimizes the asymptotic mean squared error (AMSE)[1] w.r.t. an estimate of an interested distribution $P_{\widetilde{Y}}$. As in [34, 35], the AMSE can indeed be expressed as *bias* and *variance* terms that compete with each other w.r.t. $h$. The larger $h$, the higher the bias is, while yielding a smaller variance. This then allows us to find a sweet spot on $h$ that minimizes the AMSE. Details are formally stated below.

**Theorem 1** *Assume that $\hat{y}^{(1)}, \ldots, \hat{y}^{(m)}$ are i.i.d. samples drawn from an unknown pdf $f$. Let $\widetilde{Y} := \mathbf{1}\{\widehat{Y} \geq \tau\}$. We define Kernel Density Estimator (KDE) w.r.t. $\hat{y}$ as:*

$$\hat{f}(\hat{y}) := \frac{1}{mh} \sum_{i=1}^{m} f_k \left( \frac{\hat{y} - \hat{y}^{(i)}}{h} \right) \tag{1}$$

*where $f_k$ indicates a kernel function (see Definition 4 in the main manuscript). Then the asymptotic mean squared error (AMSE) w.r.t. an estimate of $P_{\widetilde{Y}}$ can be computed as:*

$$\textit{AMSE} = \frac{h^4}{4}(f'(\tau))^2 \sigma_k^2 + \frac{\hat{P}_{\widetilde{Y}}(0)(1 - \hat{P}_{\widetilde{Y}}(0))}{m} - \frac{2hf(\tau)\int_{-\infty}^{\infty} t F_k(t) f_k(t) dt}{m} \tag{2}$$

*where $\sigma_k^2 := \int t^2 f_k(t) dt$ and $F_k(t) := \int_{-\infty}^{t} f_k(u) du$. This then yields the optimal bandwidth that minimizes the AMSE as:*

$$h^\star = \left( \frac{2f(\tau) \int t F_k(t) f_k(t) dt}{(f'(\tau))^2 \sigma_k^2} \right)^{\frac{1}{3}} \frac{1}{m^{\frac{1}{3}}}. \tag{3}$$

**Proof:** We employ the KDE (1) to derive an estimate of $P_{\widetilde{Y}}(0)$:

$$\hat{P}_{\widetilde{Y}}(0) = \int_{-\infty}^{\tau} \hat{f}(\hat{y})d\hat{y}$$

$$= \frac{1}{m}\sum_{i=1}^{m}\int_{-\infty}^{\tau} f_k\left(\frac{\hat{y}-\hat{y}^{(i)}}{h}\right)\frac{d\hat{y}}{h}$$

$$\overset{(a)}{=} \frac{1}{m}\sum_{i=1}^{m}\int_{-\infty}^{\frac{\tau-\hat{y}^{(i)}}{h}} f_k(t)\,dt \tag{4}$$

$$\overset{(b)}{=} \frac{1}{m}\sum_{i=1}^{m} F_k\left(\frac{\tau-\hat{y}^{(i)}}{h}\right)$$

where $(a)$ is due to the change of variable; and $(b)$ comes from $F_k(y) := \int_{-\infty}^{y} f_k(\hat{y})d\hat{y}$. We next compute the mean squared error (MSE) w.r.t. $\hat{P}_{\widetilde{Y}}(0)$ to represent it as the sum of the squared *bias* and *variance*:

$$\mathsf{MSE} := \mathbb{E}\left[\left(\hat{P}_{\widetilde{Y}}(0) - P_{\widetilde{Y}}(0)\right)^2\right]$$

$$= \left(\mathbb{E}\left[\hat{P}_{\widetilde{Y}}(0)\right] - P_{\widetilde{Y}}(0)\right)^2 + \mathbb{E}\left[\left(\hat{P}_{\widetilde{Y}}(0) - \mathbb{E}\left[\hat{P}_{\widetilde{Y}}(0)\right]\right)^2\right] \tag{5}$$

$$= \mathsf{bias}^2 + \mathsf{var}.$$

Let us now analyze $\mathsf{bias}$ so as to represent it as a function of $h$. To this end, we first compute the expectation of $\hat{P}_{\widetilde{Y}}(0)$:

$$\mathbb{E}\left[\hat{P}_{\widetilde{Y}}(0)\right] \overset{(a)}{=} \frac{1}{m}\sum_{i=1}^{m}\mathbb{E}\left[F_k\left(\frac{\tau-\hat{y}^{(i)}}{h}\right)\right]$$

$$\overset{(b)}{=} \mathbb{E}\left[F_k\left(\frac{\tau-\hat{Y}}{h}\right)\right] = \int_{-\infty}^{\infty} F_k\left(\frac{\tau-\hat{y}}{h}\right)f(\hat{y})d\hat{y}$$

$$\overset{(c)}{=} \int_{-\infty}^{\infty} F_k(t)f(\tau - ht)h\,dt$$

$$= -\{F_k(t)F(\tau - ht)\}|_{t=-\infty}^{t=\infty} + \int_{-\infty}^{\infty} f_k(t)F(\tau - ht)dt$$

$$\overset{(d)}{=} \int_{-\infty}^{\infty} f_k(t)F(\tau - ht)dt$$

$$\overset{(e)}{=} \int_{-\infty}^{\infty} f_k(t)\left(F(\tau) - htF'(\tau) + \frac{h^2t^2}{2}F''(\tau) + o(h^2)\right)dt$$

$$= F(\tau)\int_{-\infty}^{\infty} f_k(t)dt - hf(\tau)\int_{-\infty}^{\infty} tf_k(t)dt + \frac{h^2}{2}f'(\tau)\int_{-\infty}^{\infty} t^2 f_k(t)dt + o(h^2)$$

$$\overset{(f)}{=} F(\tau) + \frac{h^2}{2}f'(\tau)\sigma_k^2 + o(h^2)$$

$$= P_{\widetilde{Y}}(0) + \frac{h^2}{2}f'(\tau)\sigma_k^2 + o(h^2)$$

where $(a)$ comes from (4); $(b)$ is due to the i.i.d. assumption made on $\hat{y}^{(i)}$'s; $(c)$ is because of the change of variable $t = \frac{\tau-\hat{y}}{h}$; $(d)$ follows from $F_k(-\infty) = F(-\infty) = 0$ (here $F(t) := \int_{-\infty}^{t} f(u)du$); $(e)$ is because of Taylor series expansion; $(f)$ follows from the property of a kernel function: $\int_{-\infty}^{\infty} yf_k(y)dy = 0$ and our definition: $\sigma_k^2 := \int_{-\infty}^{\infty} t^2 f_k(t)dt$. Plugging this into $\mathsf{bias} := \mathbb{E}[\hat{P}_{\widetilde{Y}}(0)] - \hat{P}_{\widetilde{Y}}(0)$, we get:

$$\mathsf{bias} = \frac{h^2}{2}f'(\tau)\sigma_k^2 + o(h^2). \tag{6}$$

We next consider $\mathsf{var} := \mathbb{E}[(\hat{P}_{\tilde{Y}}(0) - \mathbb{E}[\hat{P}_{\tilde{Y}}(0)])^2]$. Using (4), we get:

$$
\begin{aligned}
\mathsf{var} &= \mathbb{E}\left[\hat{P}_{\tilde{Y}}^2(0)\right] - \left(\mathbb{E}\left[\hat{P}_{\tilde{Y}}(0)\right]\right)^2 \\
&= \mathbb{E}\left[\left(\frac{1}{m}\sum_{i=1}^{m} F_k\left(\frac{\tau - \hat{y}^{(i)}}{h}\right)\right)^2\right] - \left(\frac{1}{m}\sum_{i=1}^{m}\mathbb{E}\left[F_k\left(\frac{\tau - \hat{y}^{(i)}}{h}\right)\right]\right)^2 \\
&= \frac{1}{m^2}\mathbb{E}\left[\sum_{i=1}^{m} F_k^2\left(\frac{\tau - \hat{y}^{(i)}}{h}\right) + \sum_{i=1}^{m}\sum_{j\neq i} F_k\left(\frac{\tau - \hat{y}^{(i)}}{h}\right) F_k\left(\frac{\tau - \hat{y}^{(j)}}{h}\right)\right] \\
&\quad - \frac{1}{m^2}\left\{\sum_{i=1}^{m}\left(\mathbb{E}\left[F_k\left(\frac{\tau - \hat{y}^{(i)}}{h}\right)\right]\right)^2 + \sum_{i=1}^{m}\sum_{j\neq i}\mathbb{E}\left[F_k\left(\frac{\tau - \hat{y}^{(i)}}{h}\right)\right]\mathbb{E}\left[F_k\left(\frac{\tau - \hat{y}^{(j)}}{h}\right)\right]\right\} \\
&\overset{(a)}{=} \frac{1}{m^2}\mathbb{E}\left[\sum_{i=1}^{m} F_k^2\left(\frac{\tau - \hat{y}^{(i)}}{h}\right)\right] + \frac{1}{m^2}\sum_{i=1}^{m}\sum_{j\neq i}\mathbb{E}\left[F_k\left(\frac{\tau - \hat{y}^{(i)}}{h}\right)\right]\mathbb{E}\left[F_k\left(\frac{\tau - \hat{y}^{(j)}}{h}\right)\right] \\
&\quad - \frac{1}{m^2}\sum_{i=1}^{m}\left(\mathbb{E}\left[F_k\left(\frac{\tau - \hat{y}^{(i)}}{h}\right)\right]\right)^2 - \frac{1}{m^2}\sum_{i=1}^{m}\sum_{j\neq i}\mathbb{E}\left[F_k\left(\frac{\tau - \hat{y}^{(i)}}{h}\right)\right]\mathbb{E}\left[F_k\left(\frac{\tau - \hat{y}^{(j)}}{h}\right)\right] \\
&= \frac{1}{m^2}\sum_{i=1}^{m}\mathbb{E}\left[\left(F_k\left(\frac{\tau - \hat{Y}}{h}\right) - \mathbb{E}\left[F_k\left(\frac{\tau - \hat{Y}}{h}\right)\right]\right)^2\right] \\
&= \frac{1}{m}\left\{\mathbb{E}\left[F_k^2\left(\frac{\tau - \hat{Y}}{h}\right)\right] - \left(\mathbb{E}\left[F_k\left(\frac{\tau - \hat{Y}}{h}\right)\right]\right)^2\right\}
\end{aligned}
$$

$$(7)$$

where $(a)$ is due to the i.i.d. assumption made on $\hat{y}^{(i)}$'s. Here we manipulate the 1st term further to express it as:

$$
\begin{aligned}
\mathbb{E}\left[F_k^2\left(\frac{\tau - \hat{Y}}{h}\right)\right] &= \int_{-\infty}^{\infty} F_k^2\left(\frac{\tau - \hat{y}}{h}\right) f(\hat{y})d\hat{y} \\
&= -\int_{\infty}^{-\infty} F_k^2(t) f(\tau - ht)hdt \\
&= -\int_{-\infty}^{\infty} F_k^2(t)\, dF(\tau - ht) \\
&\overset{(a)}{=} -(F_k(t)^2 F(\tau - ht))|_{t=-\infty}^{t=\infty} + 2\int_{-\infty}^{\infty} F_k(t)f_k(t)F(\tau - ht)dt \\
&= 2\int_{-\infty}^{\infty} F_k(t)f_k(t)(F(\tau) - htF'(\tau) + o(h))dt \\
&= 2F(\tau)\int_{-\infty}^{\infty} F_k(t)f_k(t)dt - 2hF'(\tau)\int_{-\infty}^{\infty} tF_k(t)f_k(t)dy + o(h) \\
&\overset{(b)}{=} F(\tau) - 2hF'(\tau)\int_{-\infty}^{\infty} tF_k(t)f_k(t)dt + o(h) \\
&= \hat{P}_{\tilde{Y}}(0) - 2hf(\tau)\int_{-\infty}^{\infty} tF_k(t)f_k(t)dt + o(h)
\end{aligned}
$$

$$(8)$$

where $(a)$ comes from the integral by part; $(b)$ is because of $\int_{-\infty}^{\infty} F_k(t)f_k(t)dt = \frac{1}{2}$ (see below).

$$
\int_{-\infty}^{\infty} F_k(t)f_k(t)dt = \int_{-\infty}^{\infty} F_k(t)dF_k(t) = \frac{1}{2}F_k^2(t)|_{-\infty}^{\infty} = \frac{1}{2}.
$$

Plugging the above (8) into (7), we obtain:

$$\text{var} = \frac{1}{m}\left\{\mathbb{E}\left[F_k^2\left(\frac{\tau-\hat{Y}}{h}\right)\right] - \left(\mathbb{E}\left[F_k\left(\frac{\tau-\hat{Y}}{h}\right)\right]\right)^2\right\}$$

$$= \frac{1}{m}\left\{\hat{P}_{\widetilde{Y}}(0) - 2hf(\tau)\int_{-\infty}^{\infty} tF_k(t)f_k(t)dt + o(h) - \left(\mathbb{E}\left[F_k\left(\frac{\tau-\hat{Y}}{h}\right)\right]\right)^2\right\} \quad (9)$$

$$= \frac{1}{m}\left\{\hat{P}_{\widetilde{Y}}(0) - 2hf(\tau)\int_{-\infty}^{\infty} tF_k(t)f_k(t)dt - \hat{P}_{\widetilde{Y}}^2(0) + o(h)\right\}$$

where the last equality follows from taking the expectation on both sides in (4) and the fact that $\hat{y}^{(i)}$'s are i.i.d. Now this together with (6) and (5) yields:

$$\text{MSE} = \text{bias}^2 + \text{var}$$
$$= \frac{h^4}{4}(f'(\tau))^2\sigma_k^2 + \frac{\hat{P}_{\widetilde{Y}}(0)(1-\hat{P}_{\widetilde{Y}}(0))}{m} - \frac{2hf(\tau)\int_{-\infty}^{\infty} tF_k(t)f_k(t)dt}{m} + o(h^4) + o\left(\frac{h}{m}\right). \quad (10)$$

Ignoring the last two negligible terms in the above, we obtain the desired AMSE:

$$\text{AMSE} = \frac{h^4}{4}(f'(\tau))^2\sigma_k^2 + \frac{\hat{P}_{\widetilde{Y}}(0)(1-\hat{P}_{\widetilde{Y}}(0))}{m} - \frac{2hf(\tau)\int_{-\infty}^{\infty} tF_k(t)f_k(t)dt}{m}. \quad (11)$$

One can also readily verify that the optimal bandwidth that minimizes the AMSE is:

$$h^\star = \left(\cdot\frac{2f(\tau)\int yF_k(y)f_k(y)dy}{(f'(\tau))^2\sigma_k^2}\right)^{\frac{1}{3}}\frac{1}{m^{\frac{1}{3}}}.$$

∎

## 3   Explicit Gradient Formulas

Our KDE-based approach enables us to express the fairness-related regularization term as a function of model parameters. Hence, we can train a fair classifier using a standard machine learning library such as PyTorch [26] that comes with an *autograd* function. Nonetheless, providing an explicit gradient formula would help readers to better understand our approach. To this end, we first calculate the gradient of the model output w.r.t. the model parameters:

$$\nabla_w \hat{y}^{(i)}. \quad (12)$$

Then using this gradient we provide the closed forms of the gradients of DDP and DEO when we employ a linear network or a 2-layer NN as a classifier model.

### 3.1   The gradient of the model output

#### 3.1.1   A linear model

Under a linear classifier, we assume that the model parameters consist of a weight $W$ and a bias $b$:

$$w := (W, b) \quad (13)$$

Then the output of the classifier can be calculated as:

$$\hat{y}^{(i)} = \sigma(W^T\tilde{x}^{(i)} + b) \quad (14)$$

where $\sigma(\cdot)$ is a sigmoid function and $\tilde{x}^{(i)} := (x^{(i)}, z^{(i)})$. Now we can compute the gradient of $\hat{y}^{(i)}$:

$$\nabla_W \hat{y}^{(i)} = \sigma(W^T\tilde{x}^{(i)} + b)\left(1 - \sigma(W^T\tilde{x}^{(i)} + b)\right)\tilde{x}^{(i)} \quad (15)$$

$$\nabla_b \hat{y}^{(i)} = \sigma(W^T\tilde{x}^{(i)} + b)\left(1 - \sigma(W^T\tilde{x}^{(i)} + b)\right). \quad (16)$$

### 3.1.2 A 2-layer NN classifier

Under a 2-layer NN, model parameters are:

$$w := (W^{[1]}, b^{[1]}, W^{[2]}, b^{[2]}). \tag{17}$$

where $W^{[j]}$ and $b^{[j]}$ are a weight matrix and a bias vector at the $j$-th layer. Then the forward pass for the $i$-th input data $\tilde{x}^{(i)}$ is:

$$o^{[1](i)} = W^{[1]}\tilde{x}^{(i)} + b^{[1]} \ (\in \mathbb{R}^{d_1}), \tag{18}$$

$$a^{[1](i)} = \sigma^{[1]}(o^{[1](i)}) \ (\in \mathbb{R}^{d_1}), \tag{19}$$

$$o^{[2](i)} = W^{[2]}a^{[1](i)} + b^{[2]} \ (\in \mathbb{R}), \tag{20}$$

$$\hat{y}^{(i)} = a^{[2](i)} = \sigma^{[2]}(o^{[2](i)}) \ (\in \mathbb{R}) \tag{21}$$

where $o^{[j]}$ refers to the intermediate output in the $j$-th layer, and $a^{[j]}$ represents the output of activation function $\sigma^{[j]}(\cdot)$ in the $j$-th layer. Let $\sigma^{[1]}$ be a ReLU function and $\sigma^{[2]}$ be a sigmoid function as we are interested in a binary classification setting. Applying the back-propagation rule on the model parameters $(W^{[1]}, b^{[1]}, W^{[2]}, b^{[2]})$, we get:

$$\nabla_{W^{[2]}}\hat{y}^{(i)} = \sigma^{[2]}(o^{[2](i)})\left(1 - \sigma^{[2]}(o^{[2](i)})\right)a^{[1](i)T} \tag{22}$$

$$\nabla_{b^{[2]}}\hat{y}^{(i)} = \sigma^{[2]}(o^{[2](i)})\left(1 - \sigma^{[2]}(o^{[2](i)})\right) \tag{23}$$

$$\nabla_{W^{[1]}}\hat{y}^{(i)} = \sigma^{[2]}(o^{[2](i)})\left(1 - \sigma^{[2]}(o^{[2](i)})\right)W^{[2]T}.*\mathbf{1}\{o^{[1](i)} \geq 0\}\tilde{x}^{(i)T} \tag{24}$$

$$\nabla_{b^{[1]}}\hat{y}^{(i)} = \sigma^{[2]}(o^{[2](i)})\left(1 - \sigma^{[2]}(o^{[2](i)})\right)W^{[2]T}.*\mathbf{1}\{o^{[1](i)} \geq 0\} \tag{25}$$

where $.*$ and $\mathbf{1}(\cdot)$ represent an element-wise multiplication and an indicator function, respectively.

### 3.2 The gradient of DDP

In our approach, we employ a Gaussian kernel function:

$$f_k(\hat{y}) := \frac{1}{\sqrt{2\pi}}\exp\left(-\frac{\hat{y}^2}{2}\right) \tag{26}$$

and the approximated cdf of $f_k(\hat{y})$:

$$F_k(\hat{y}) := \int_{\hat{y}}^{\infty} f_k(y)dy \approx e^{-a\hat{y}^2 - b\hat{y} - c} \tag{27}$$

where $(a, b, c) = (0.4920, 0.2887, 1.1893)$. Let us recall the formula of the approximated gradient of DDP in the main body of the paper. We first estimate $f_{\hat{Y}|Z}(\hat{y}|z)$ using the KDE:

$$\hat{f}_{\hat{Y}|Z}(\hat{y}|z) = \frac{1}{m_z h}\sum_{i \in I_z} f_k\left(\frac{\hat{y} - \hat{y}^{(i)}}{h}\right) \tag{28}$$

where $I_z := \{i : z^{(i)} = z\}$ and $m_z := |I_z|$. This together with $\widetilde{Y} := \mathbf{1}\{Y \geq \tau\}$ gives:

$$\hat{P}_{\widetilde{Y}|Z}(1|z) = \int_{\tau}^{\infty} \hat{f}_{\hat{Y}|Z}(\hat{y}|z)d\hat{y} = \frac{1}{m_z}\sum_{i \in I_z} F_k\left(\frac{\tau - \hat{y}^{(i)}}{h}\right)$$

where $F_k(\hat{y}) := \int_{\hat{y}}^{\infty} f_k(y)dy$.

**Proposition 1** *Since $f_k(\hat{y})$ is continuous and each $\hat{y}^{(i)}$ is a differentiable function w.r.t. $w$, $\hat{P}_{\widetilde{Y}|Z}$ is also differentiable. Using the chain rule, one can then compute its gradient as:*

$$\nabla_w \hat{P}_{\widetilde{Y}|Z}(1|z) = \frac{1}{m_z h}\sum_{i \in I_z} f_k\left(\frac{\tau - \hat{y}^{(i)}}{h}\right) \cdot \nabla_w \hat{y}^{(i)}. \tag{29}$$

Then the approximated DDP is:

$$\mathsf{DDP} \approx \sum_{z \in \mathcal{Z}} \left| \hat{P}_{\widetilde{Y}|Z}(1|z) - \hat{P}_{\widetilde{Y}}(1) \right| \text{ and } \hat{P}_{\widetilde{Y}}(1) = \sum_{z \in \mathcal{Z}} \frac{m_z}{m} \hat{P}_{\widetilde{Y}|Z}(1|z). \tag{30}$$

For tractability of the non-differentiable absolute function $|\cdot|$, we employ the Huber loss [13]:

$$\mathsf{DDP} \approx \sum_{z \in \mathcal{Z}} H_\delta \left( \hat{P}_{\widetilde{Y}|Z}(1|z) - \hat{P}_{\widetilde{Y}}(1) \right) \text{ where } H_\delta(x) := \begin{cases} \frac{1}{2}x^2 & \text{for } |x| \leq \delta; \\ \delta(|x| - \frac{1}{2}\delta) & \text{otherwise.} \end{cases}$$

This together with (29) and (30) yields an approximation of the gradient of $\mathsf{DDP}$:

$$\nabla_w \mathsf{DDP} \approx \sum_{z \in \mathcal{Z}} H'_\delta \left( \hat{P}_{\widetilde{Y}|Z}(1|z) - \hat{P}_{\widetilde{Y}}(1) \right) \cdot \nabla_w \left( \hat{P}_{\widetilde{Y}|Z}(1|z) - \hat{P}_{\widetilde{Y}}(1) \right) \tag{31}$$

where $H'_\delta(x) := \begin{cases} x & \text{for } |x| \leq \delta; \\ \delta & \text{for } x > \delta; \\ -\delta & \text{for } x < -\delta. \end{cases}$

By substituting the gradient of the model output computed in subsection (3.1) into (29), we can obtain the closed-form of $\nabla_w \mathsf{DDP}$ under a linear network or a 2-layer NN.

### 3.3 The gradient of DEO

Taking the KDE approach, similarly we obtain:

$$\hat{P}_{\widetilde{Y}|Z,Y}(1|z,y) = \frac{1}{m_{zy}} \sum_{i \in I_{zy}} F_k \left( \frac{\tau - \hat{y}^{(i)}}{h} \right) \tag{32}$$

where $I_{zy} := \{i : z^{(i)} = z, y^{(i)} = y\}$ and $m_{zy} := |I_{zy}|$. We can then compute the gradient w.r.t. $w$:

$$\nabla_w \hat{P}_{\widetilde{Y}|Z,Y}(1|z,y) = \frac{1}{m_{zy}h} \sum_{i \in I_{zy}} f_k \left( \frac{\tau - \hat{y}^{(i)}}{h} \right) \cdot \nabla_w \hat{y}^{(i)}. \tag{33}$$

Again using the KDE together with the Huber loss, we approximate:

$$\mathsf{DEO} \approx \sum_{y \in \{0,1\}} \sum_{z \in \mathcal{Z}} H_\delta \left( \hat{P}_{\widetilde{Y}|Z,Y}(1|z,y) - \hat{P}_{\widetilde{Y}|Y}(1|y) \right) \tag{34}$$

where $\hat{P}_{\widetilde{Y}|Y}(1|y) = \sum_{z \in \mathcal{Z}} \frac{m_{zy}}{m_y} \hat{P}_{\widetilde{Y}|Z,Y}(1|z,y)$ and $m_y := |\{i : y^{(i)} = y\}|$. This then yields:

$$\nabla_w \mathsf{DEO} \approx \sum_{y \in \{0,1\}} \sum_{z \in \mathcal{Z}} H'_\delta \left( \hat{P}_{\widetilde{Y}|Z,Y}(1|z,y) - \hat{P}_{\widetilde{Y}|Y}(1|y) \right) \cdot \nabla_w \left( \hat{P}_{\widetilde{Y}|Z,Y}(1|z,y) - \hat{P}_{\widetilde{Y}|Y}(1|y) \right).$$
$$\tag{35}$$

By substituting the gradient of the model output computed in subsection (3.1) into (33), we can obtain the closed-form of $\nabla_w \mathsf{DEO}$ under a linear network or a 2-layer NN.

## 4 Synthetic Data Experiments

We provide details on how unfair synthetic datasets are generated and demonstrate accuracy-fairness tradeoff performance.

### 4.1 Binary classification

There are two *unfair* synthetic datasets that we used in our paper: one based on the Moon dataset [10] and the other for 3-way classification. For the former, we first get a dataset $\{(x^{(i)}, y^{(i)}\}_{i=1}^{15,000}$ using the make_moons function of Scikit-learn [10], an open source machine learning library. For indices of positive examples ($y^{(i)} = 1$), we generate $z^{(i)}$ as per a multinomial distribution $\mathsf{Multinomial}(0.5, 0.2, 0.3)$ with a single trial; otherwise, $z^{(i)}$ follows $\mathsf{Multnomial}(0.2, 0.3, 0.5)$ with a single trial. Note that the conditional probabilities $\Pr(Y = 1|Z = z)$ are distinct across different demographics $z$. This way, we could generate a balanced yet unfair dataset in which $\Pr(Y = 1|Z = 0) \approx 0.720$, $\Pr(Y = 1|Z = 1) \approx 0.396$, and $\Pr(Y = 1|Z = 2) \approx 0.379$ while respecting $Y \sim \mathsf{Bern}(0.5)$. The generated dataset is visualized in Fig. 2 of our main paper.

## 4.2 Multiclass classification

The other synthetic dataset that we used for the multiclass classification is generated as follows: we first create a mixture of three Gaussians $\text{Normal}((2,2), I_2)$, $\text{Normal}((-2,2), I_2)$, and $\text{Normal}((0,-2), I_2)$ with label values of 0, 1, and 2, respectively. Then for examples with $(y^{(i)} = 0)$, we assign a sensitive attribute $z$ as per $\text{Bern}(0.8)$, while we use $\text{Bern}(0.2)$ for the remaining examples. This way, we could get an unfair dataset in which $\Pr(Y = 0|Z = 0) \approx 0.102$, $\Pr(Y = 0|Z = 1) \approx 0.665$, $\Pr(Y = 1|Z = 0) \approx 0.450$, $\Pr(Y = 1|Z = 1) \approx 0.166$, $\Pr(Y = 2|Z = 0) \approx 0.448$, and $\Pr(Y = 2|Z = 1) \approx 0.169$. The generated dataset is visualized in Fig. 5 of our main paper.

## 5 Real Data Experiments

We now present accuracy-fairness tradeoff performances on two additional real benchmark datasets: Law School Admissions [36] and Credit Card Default [6, 39]. To compare the computational complexity of our algorithm with others, we also include a comprehensive table presenting the running times of all the baseline algorithms on all benchmark datasets.

### 5.1 Performances on Law School Admissions [36] and Credit Card Default [6, 39]

Figure 1: (Left) Accuracy-fairness tradeoff w.r.t. Demographic Parity evaluated on the Law School Admissions dataset; (Right) Equalized Odds counterpart.

Figure 2: (Left) Accuracy-fairness tradeoff w.r.t. Demographic Parity evaluated on the Credit Card Default dataset; (Right) Equalized Odds counterpart.

Fig. 2 shows accuracy-fairness tradeoff evaluated on Credit Card Default w.r.t. DDP and DEO. Each point corresponds to a particular tuning knob and it represents an average value over 5 trials with different random seeds. Off-the-scale curves for low-performance baselines are not shown. We employ a 2-layer NN with 32 hidden nodes for all baselines but logistic regression for Zafar et al. [42] and Narasimhan [25]. For Zhang et al. [44], we use a linear discriminator that turns out to yield more stable training. The tradeoff performance of ours w.r.t. DDP shows respectful margins compared to other baselines but Zhang et al. [44] also yields relatively good performance in respecting DEO. However, note that ours achieves slightly better tradeoff performance in the non-perfect fairness regime that favors the accuracy performance relative to the fairness performance. Fig. 1 presents the

same performances as in Fig. 2, yet on Law School Admissions dataset. The only distinction in an experimental setting is that the 2-layer NN classifiers now have 16 hidden nodes.

## 5.2 Training Details

Below we leave two tables: one that contains hyperparameters for our approach used in real data experiments (See Table 1), and the other that includes the running times for all baseline algorithms under the considered benchmark datasets (See Table 2).

Table 1: Hyperparameters used for real data experiments. Each entry includes hyperparameters for experiments w.r.t. DDP/DEO

| Datasets | Law School Admissions | Adult Census | Credit Card Default | COMPAS |
|---|---|---|---|---|
| $h$ (bandwidth of KDE) | 0.1 / 0.1 | 0.1 / 0.1 | 0.1 / 0.1 | 0.1 / 0.1 |
| $\delta$ (of Huber function) | 1.0 / 1.0 | 1.0 / 1.0 | 1.0 / 1.0 | 1.0 / 1.0 |
| $\tau$ (threshold for hard decision of $\hat{Y}$) | 0.5 / 0.5 | 0.5 / 0.5 | 0.5 / 0.5 | 0.5 / 0.5 |
| Batch size | 2048 / 2048 | 512 / 512 | 2048 / 2048 | 2048 / 2048 |
| $(\beta_1, \beta_2)$ (for Adam optimizer) | (0.9, 0.999) / (0.9, 0.999) | (0.9, 0.999) / (0.9, 0.999) | (0.9, 0.999) / (0.9, 0.999) | (0.9, 0.999) / (0.9, 0.999) |
| Learning rate | 2e-4 / 2e-4 | 1e-1 / 1e-1 | 5e-4 / 5e-4 | 5e-4 / 2e-4 |
| Exponential decay factor (for LR scheduler) | None / None | 0.98 / 0.96 | None / None | None / None |
| Number of epochs | 200 / 200 | 200 / 200 | 300 / 300 | 500 / 500 |

Table 2: The amount of time (in seconds) to obtain a single point on the accuracy-DDP tradeoff curve. Each number indicates an average of 5 trials for a specific tuning knob.

| Dataset | Hardt et al. [12] | Narasimhan [25] | Zafar et al. [42] | Zhang et al. [44] | Agarwal et al. [1] | Proposed |
|---|---|---|---|---|---|---|
| Law School Admissions | 16.7 | 58.7 | 13.6 | 79.1 | 423.9 | 157.0 |
| Adult Census | 11.1 | 43.5 | 13.1 | 77.8 | 334.1 | 111.1 |
| Credit Card Default | 12.9 | 16.3 | 13.9 | 94.8 | 353.5 | 75.5 |
| COMPAS | 10.9 | 4.0 | 13.1 | 51.7 | 171.1 | 25.3 |

Table 2 emphasizes a computational challenge that arises in Agarwal et al. [1], a baseline algorithm that also shows outstanding performances across different datasets. It turns out that proposed algorithm takes only about 15∼37% amount of time, as compared to [1], while being comparable to Zhang et al. [44]. We observe similar tendencies for experiments w.r.t. DEO. All the algorithms are trained until a training loss converges while other hyperparameters are chosen so as to achieve the best tradeoff.

Experiments on synthetic and four real benchmark datasets demonstrate that the proposed algorithm consistently achieves near best accuracy-fairness tradeoff performance with a reasonable amount of computational time.

## Footnotes

[1] Here the AMSE refers to an approximated MSE which neglects some higher-order terms w.r.t. $h$ that arise in applying Taylor series expansions.