[Reviews · NeurIPS 2020]

Review 1

Summary and Contributions: Update: I'm raising my score one point in response to the author rebuttal. The paper proposes a kernel method to estimate the PMF of the hard predictions w.r.t. the model parameters.

Strengths: Common group fairness metrics are computed using hard predictions. The key benefit of the proposed method is that (approximations to) fairness metrics relying on these hard computations can be used by automatic differentiation software.

Weaknesses: The authors acknowledge that kernel methods may not scale to high dimensional problems. Furthermore the method as stated is limited to classification problems with binary targets. The experimental section is somewhat unsatisfactory (will describe in further detail below). Also the benefit of the kernel method relative to alternative approaches seems to be different in the motivation than it is in the experiments, which makes the paper feel somewhat disjointed (discussed below).

Correctness: I believe so.

Clarity: There are some typos. Also I think the introduction is quite vague and could be improved by either adding citations to general descriptions of recent trends, or focusing more on this hard/soft prediction issue as a key motivation for the proposed method.

Relation to Prior Work: The discussion of the adversarially learned literature is somewhat incomplete. In addition to the work by Zhang the following papers should be discussed * https://arxiv.org/abs/1511.05897 * https://arxiv.org/abs/1511.00830 (a VAE counterpart to the GAN appraoch) * https://arxiv.org/abs/1802.06309

Reproducibility: Yes

Additional Feedback: I think the overall idea is somewhat intriguing, and agree that the distinction between hard prediction and model-assigned soft probabilities is typically glossed over in the literature. But sections 3 and 4 seems somewhat disconnected to me, since this differentiability issues does not come across strongly in the experiments (instead the instability of baselines due to adversarial training is emphasized). I also think that laying out the extending to multi-class classification would help the paper a lot (to save space experimental details could be moved to the appendix) About the experiments, there are several improvements to be made. I have a concern about whether the baselines were substantially tuned, since there is no discussion of how hyperparameters such as learning rates were selected for the baselines. I also didn't understand the choice of a narrow MLP architecture (14 hidden units per layer) for the neural net methods. It seems that the main claim in section 3 is that the kernel method will help by making the objective differentiable, but in section 4 the claim is that the kernel method is more stable than competing methods. Is there an ablation (even in a controlled synthetic setting) that could show that the differentiability of the kernel method makes a meaningful difference in the performance? I don't think it makes sense to specifically name the synthetic data as coming from "Black"/"white" racial groups since this data is entirely synthetic (use "majority"/"minority" instead). typos: S1 - "that serves to estimate probability distribution" -> "that serves to estimate a probability distribution" S4 - "...a single-layer linear class." -> "...a single-layer linear classifier" "spreadness" is an awkward word; use "variability"


Review 2

Summary and Contributions: The paper proposes differentiable fairness (demographic parity and equalised odds) regularisors for classifiers based on minimising the cross-entropy loss. The fairness regularisor is computed by estimating the conditional distribution of the classifier's probability scores given protected attributes (and outcome in the case of demographic parity) via kernel density estimation. They claim that this approach improves on other approaches to fair classification because the regularisor is based on direct estimation of the conditional distributions on which the fairness metrics are based, rather than optimising a proxy such as the covariance. They show empirically that their approach achieves a better fairness/accuracy trade-off than some alternate approaches on simulated data and that it is comparable to Agarwal et al on real datasets. They claim their approach has better runtime performance than Agarwal et al.

Strengths: The proposed approach is simple, straightforward to implement and the authors provide code. The experiments provided are suggestive that this approach may be competitive with more complex approaches.

Weaknesses: The major weakness of this paper is the lack of theory to back the claim that the approach will outperform existing methods. If it is not possible to prove improved performance due to the non-convexity of the problem, this should be acknowledged and discussed. To make a claim about improved performance without theoretical results, the empirical experiments would need to be comprehensive and to explore cases where the method might intuitively be expected to fail, as well as settings where we would expect it to succeed. Unfortunately, the empirical experiments fail to separate improvement due to the alternate regularisor from improvement due to the use of a more flexible model. It is hardly surprising that that a layer NN outperforms logistic regression on synthetic data that is not linearly separable. They state that that their approach is faster than that of Agarwal et al, but do not attempt any time complexity analysis, theoretical or empirical(ie by plotting time vs number of data points), that would demonstrate that this improvement is due to the algorithm as opposed to implementation details.

Correctness: The claims of the paper (regarding improved performance) need to be more formally stated in order for their correctness to be properly assessed. The empirical methodology appears reasonable, however the breadth of experiments run is not sufficient to provide strong evidence for the claims made.

Clarity: Overall the paper is readable, but the problem statement and contribution claims need to be stated more clearly (formally). The problem statement should clearly state the class of problems you are attempting to solve - eg fair classification with respect to DP or EO with discrete (binary?) protected attributes. The discussion of the regularised classifier (starting line 110) belongs in the proposed approach section. Statements such as the KDE trick "allows us to faithfully quantify fairness metrics" should be more formally stated or omitted.

Relation to Prior Work: No, the paper focuses discussion and comparison only three competing in-processing approaches: Agarwal et al, Zhang et al and Zafar et al. They fail to discuss other approaches to directly estimating and regularising independence based measures in particular approaches based on information theory, eg see: Wasserstein Fair Classification (Deepmind), Fairness-Aware Classifier with Prejudice Remover Regularizer, Fairness Measures for Regression via Probabilistic Classification

Reproducibility: Yes

Additional Feedback: The authors have addressed my points of concern in their response. With the experiments the authors propose to complete, and perhaps some additional experiments testing how performance degrades as the number of protected attributes (and thus dimension of the density estimate) increases, this could be a strong paper. The approach is straightforward, applicable to many model classes and could be a practical way of encoding fairness concerns in real AI systems. Unfortunately, given the extent of the proposed changes and the fact that the experiments currently in the paper fall short of demonstrating the key claim of improved performance I don't support accepting this submission this time around.


Review 3

Summary and Contributions: Update: After reading all reviews and rebuttals, I agree that the exps in current version is incomplete. My major concern is that the datasets are too simply. During the rebuttal, the author said that they conducted exps on complex datasets and observed the same trends. I believe the author's comments. However at the current stage, there are multiple exps missing. The current submission is not ready. I would lower the score to 5. This paper learns a fair classifier. Specifically, the input to the classifier is data with both ordinary and sensitive (e.g. gender, races) information. The model needs to learn a classifier that is irrelevant to the sensitive information. The criterion of determining whether the classifier is relevant to the sensitive information is Demographic Parity (DDP) and Equalized Odds (DEO). Directly optimizing both criterions is not easy. Previous works use proxy regularizer to enforce those two criterion. This paper proposed to directly estimate the DDP and DEO using kernel density estimation. The proposed approach achieves good performance when the prediction is binary.

Strengths: Strength: 1. The paper is well-written and easy to follow. 2. The contribution is clear that the paper uses KDE to directly estimate the DDP and DEO. KDE can also be differentiable. Therefore the gradient descent can be used to train the model. 3. Comparing with other approach, when the decision space is binary, the proposed approach shows stability in training and better performance.

Weaknesses: Weakness: 1. As mentioned in Remark 1, the KDE would requires more data when the classifier space is beyond binary. However for other approach, Zhang et. al.[44], this seems not a problem. I wonder whether it is possible to show the comparison of the performance on datasets that the decision space is beyond binary? 2. Is it possible to show the relation between the size of decision space and the amount of data required for KDE? 3. Experiments: a. My first concerns in the experiment section is whether the dataset is challenging enough? The setting of all real-world datasets seems very simplified. Specifically, the decision space is binary and only ONE attribute is considered as sensitive information. Therefore, I wonder whether the proposed approach would work in a more complex setting where more attributes are considered as sensitive information. b. I have another concerns about the potential impact of the proposed approach. Specifically, I wonder would this approach be generalized to other tasks? Comparing with the experiments in [44], the approach in [44] seems could be applied into a more broad use case, which is the de-biasing on word embedding. However, it seems non-trivial for the proposed approach to generalize beyond the binary decision problem. c. Comparing with [44], apart from stabilized training, what is the advantage of the proposed approach over [44]. Performance wise, it seems the proposed approach achieves similar results as [44] in Credit Card dataset and COMPAS dataset. Computational time wise, the proposed approach is actually slower than [44] on Law School dataset and adult census datasets. Application wise, it seems [44] could be applied to a broader application domain than the proposed approach.

Correctness: The method is sound and clear.

Clarity: The paper is well written. The whole storyline and approach is easy to follow. The task is clearly presented.

Relation to Prior Work: The relation to previous work is clearly discussed.

Reproducibility: Yes

Additional Feedback: Although I am not very familiar with this area, I feel the contribution of the proposed approach is important. Specifically, enabling end-to-end training from DEO and DDP is an important contribution. However I have concerns with the following two points. 1. It seems the applications that the proposed approach could be applied are pretty limited (weakness 3.b). 2. Comparing with [44], what is the advantage of the proposed approach? (weakness 3.c)


Review 4

Summary and Contributions: The paper uses kernel density estimation to directly quantify measures of fairness without using a proxy. It achieves good accuracy in measuring the probability distributions used to compute the fairness measure, and the fairness measure can be directly optimized via gradient descent. Empirically, this method achieves good accuracy-fairness tradeoff and improves training stability when compared to prior methods.

Strengths: Based on the comparison to prior work described in the paper, I believe that the contribution is novel, in particular the direct computation of an interested fairness measure and the ability to directly optimize it, without relying on fairness proxies such as a covariance function. Empirical evaluations are performed along various axes (e.g. accuracy-fairness tradeoff, training stability, robustness to hyper-parameters) to demonstrate the utility of the proposed approach. I also believe that this work is highly relevant to the NeurIPS community.

Weaknesses: In high dimensions, the KDE approach gives an inaccurate distribution estimate without an exponential number of samples with respect to the number of dimensions. The authors note that this is not the case when using a binary classifier, but this means that the KDE approach is limited (with respect to its ability to be used in practice) to the binary classification setting. An empirical evaluation of the accuracy of the pmf distribution estimate Pr(\tilde{Y}= 1) is not present.

Correctness: To the best of my knowledge, the claims and method are correct. In Figure 1, I am curious about which dataset was used in computing the pdf estimates of \hat{Y} and pmf estimates of \tilde{Y}. Also, is there a way to quantify the robustness or compare robustness between the pdf and pmf estimates in a quantitative way? Finally, while the right table in Figure 1 demonstrates the robustness of the estimated pmf of \tilde{Y} against various h's, it is not clear to me how this correlates with accuracy.

Clarity: Yes, as someone without a background in fairness, I found the paper both instructive in discussing fairness measures and prior approaches, as well as clear in describing the method and its benefits. The application of a kernel density estimator to construct the loss function to be added as a regularizer was well motivated and explained. Overall, the paper strikes a good balance between conciseness and explanation. The appendix is also well structured and helpful.

Relation to Prior Work: Yes. It is discussed how this work differs from contributions which use a fairness proxy and works which use an adversarial learning framework to ensure that predictions are made independently of sensitive attributes. The key differences from those works are that the fairness measures are directly quantified, so fairness measures are well-respected, and no adversarial optimization is required, therefore improving training stability.

Reproducibility: Yes

Additional Feedback: A note: Fairness in machine learning is not my area of expertise, and therefore, I am not at all familiar with the related work. However, I was able to understand the paper and the approach that was described. A naive question: If sensitive attributes are known apriori, why not simply remove them from the dataset rather than enforcing a fairness constraint based on independence of the prediction and the sensitive attribute? (This is briefly mentioned in lines 88-90.) Wouldn't this automatically ensure fairness under both the Demographic Parity and Equalized Odds definitions? Update (post rebuttal): I am lowering my score to a 5 based on the need for a more comprehensive experimental evaluation, such as having experimental results with multi-class classifiers and results on more complex datasets.

[Author Response · NeurIPS 2020]

**To all reviewers**  We would like to thank the reviewers for their thoughtful comments and useful suggestions, which
helped us improve the manuscript. Below we provide point-by-point responses.

**To Reviewer 1**
*[R1-1] (Extension to multi-class classifiers)* As per your great suggestion, we will present a multi-class classifier
counterpart that we developed yet not included in the current draft. The idea is to make a binary hard decision
*individually for each element* in the softmax output, and then to estimate the pmf of each hard decision via KDE. For a
3-way classification synthetic dataset (3 Gaussian mixture), this extended method offers respectful improvements over
[44] - trends of the gains are similar to those in Figs. 2 and 3. In a revision, we will provide details on our extension
together with the experimental results while moving experimental details to the appendix as suggested.
*[R1-2] (Hyperparameter tuning)* (a) Yes, we exhaustively searched hyperparameters for the baselines. For instance, the
learning rate was best-picked among several log-scaled candidates (b) Similarly the narrow MLP was chosen as a result
of search - we found the depth and width do not affect too much in performance at least for the considered benchmark
datasets. We will clarify these in a revision.
*[R1-3] (Motivation of the kernel method and writing-flow)* (a) In fact, the stability is also a key issue that we wanted
to highlight, although it is not well motivated in the current introduction. We will rewrite the introduction to better
balance the issues while citing relevant papers suggested. (b) Yes, the hard/soft decision issue is a key motivation,
and the relevant insight allowed us to use the kernel method powerfully and beneficially. For a smooth logical flow,
we will re-balance Secs. 3 & 4 so that the issue is strongly emphasized together with an ablation study (w/ and w/o
differentiability) in a synthetic setting (as suggested).
*[R1-4] (Other comments)* Thanks for your suggestion of the "majority"/"minority" naming as well as pointing out typos.
We will fix them.

**To Reviewer 2**
*[R2-1] (Lack of theory)* We fully agree that the theory is missing for the main claim re. improved performance. Yes,
the analysis was not that simple due to the non-convexity of the problem. We will acknowledge this with a proper
discussion in a revision.
*[R2-2] (Comprehensive experiments for supporting performance gains?)* As per your great suggestion, we will conduct
an ablation study in which one may be able to separate improvement due to our regularizer from that due to the use of a
more flexible model. Specifically we will compare ours to a kernel SVM by Zafar et al. and include this result in a
revision.
*[R2-3] (Complexity comparison w.r.t. Agarwal et al)* As you may guess, the theoretical complexity analysis of our
algorithm was not done although we empirically demonstrated that ours exhibits lower complexity relative to Agarwal
et al. (requiring multiple rounds of training); see Table 1 in supplementary. Instead we will provide in-depth empirical
analysis by plotting the running time as a function of the number of data points, as suggested.
*[R2-4] (Problem statement & organization)* As per your suggestion, we will make the problem statement clearer and
more formal, as well as move the regularizer part into the "Proposed Approach" section.
*[R2-5] (Relation to prior work & additional feedback)* Thanks for pointing out the approaches (Wasserstein Fair
Classifier etc) that directly estimate fairness measures via information theory. We will cite them with enough discussion.
Also we will include the kernel SVM by Zafar et al. and Agarwal et al. in the synthetic setting.

**To Reviewer 3**
*[R3-1] (Extension to multi-class settings and performance comparison)* As mentioned in response to [R1-1], we actually
developed a generalized kernel method that is applicable to multi-class settings. As per your suggestion, we also made
performance comparison on a 3-way classification synthetic dataset, observing similar gains as those exhibited in Figs.
2 and 3. We will discuss all of these in a revision.
*[R3-2] (Multiple sensitive attributes)* Yes, our approach works well for the complex setting. We now conducted
experiments on one such setting (AdultCensus with two sensitive attributes: race, gender), observing similar performance
improvements, as those in Figs. 2 and 3. We will include the results in a revision.
*[R3-3] (Comparison to [44] in many aspects)* While our approach offers key benefits in training stability and tradeoff
performance, we do agree that [44] is more flexible in terms of application domains. For a fair comparison, we will
summarize pros-&-cons of our approach relative to [44] in many aspects.

**To Reviewer 4**
*[R4-1] (Dataset set in Fig. 1 and accuracy of the pmf estimate)* We employed a Gaussian mixture: $0.3 \cdot \mathcal{N}(0.37, 0.0055) +$
$0.7 \cdot \mathcal{N}(0.74, 0.0055)$. Here the true probability is around 0.7, and this is very close to the pmf estimates in Fig. 1(Right).
We will provide this in a revision.
*[R4-2] (Robustness quantification and its relation with accuracy)* One way of quantification is to compute the variance
of the pmf estimates over different $h$'s. We will mention this in a revision while plotting accuracy as a function of $h$.
*[R4-3] (Removal of sensitive attributes?)* Yes, that is one natural trial. However, such removal does not ensure fairness
especially when $X$ is correlated with $Z$. Please see [42] for details.

[Meta-Review · NeurIPS 2020]

The paper proposes a simple but rather practical approach to estimate statistical fairness notions without relying on a proxy, in contrast to several prior work. The proposed approach relies on Kernel Density Estimation (KDE), which allows to compute the gradient of the fairness notion with respect to the model parameters in close form, easing the learning procedure of a fair classifier. As a result, he proposed approach leads to a better fairness accuracy trade-off than competing methods in several datasets. In particular, the experiments show that the proposed approach outperforms prior work relying on fairness proxies, and leads more stable results that approaches that rely on adversarial training top trade-off fairness and accuracy. In fact, the empirical results are comparable to the ones provided by Agarwal et al. (2018), whose solution provide theoretical guarantees but comes at a high computational cost. Although there exists extensive literature on solving the fair classification problem, the empirical results show the efficacy of KDE in this context. Moreover, this work may also trigger future work, where the KDE may be used to solve the highly non-convex original constraint optimization problem (rather than the regularized problem), navigate the Pareto frontier between fairness and accuracy, or solve the problem of fair decision making, e.g., under selective labels where classification approaches are not optimal anymore. The reviewers agree on the simplicity and potential usefulness of the proposed approach--e.g., R1 mentions that "The approach is straightforward, applicable to many model classes and could be a practical way of encoding fairness concerns in real AI systems." The authors properly responded in the rebuttal to the major points raised by the reviewers, being the main concern that remains the need of additional experiments. In the rebuttal, the authors do already mention their extension and synthetic experiments to the multi-class settings and multiple sensitive attributes, as well as their plan to conduct an ablation study and more detail comparison with [44]. Based on such detailed experiment description, I believe that this is an easy fix for the camera-ready, and I encourage the authors to thoroughly incorporate all the reviewers' feedback in the revised version of their paper, as it will significantly improve the potential impact of the paper.